# Efficient Inference With Model Cascades

**Luzian Lebovitz**     *luzian.lebovitz@pbl.ee.ethz.ch*
*Department of Electrical Engineering & Information Technology*
*ETH Zurich*

**Lukas Cavigelli**     *lukas.cavigelli@huawei.com*
*Computing Systems Lab*
*Huawei Technologies*

**Michele Magno**     *michele.magno@pbl.ee.ethz.ch*
*Department of Electrical Engineering & Information Technology*
*ETH Zurich*

**Lorenz Müller**     *lorenz.mueller@huawei.com*
*Computing Systems Lab*
*Huawei Technologies*

**Reviewed on OpenReview:** *https://openreview.net/forum?id=obB415rg8q*

## Abstract

State-of-the-art deep learning models are becoming ever larger. However, many practical applications are constrained by the cost of inference. Cascades of pretrained models with conditional execution address these requirements based on the intuition that some inputs are easy enough that they can be processed correctly by a smaller model allowing for an early exit. If the smaller model is not sufficiently confident in its prediction, the input is passed on to a larger model. The selection of the confidence threshold allows to trade off computational cost against accuracy. In this work we explore the effective design of model cascades, thoroughly evaluate the impact on the accuracy-efficiency trade-off, and provide a reproducible state-of-the-art baseline that is currently missing for related research. We demonstrate that model cascades dominate the ImageNet Pareto front already with 2-model cascades, achieving an average reduction in compute effort at equal accuracy of almost $3.1\times$ above 86% and more than $1.9\times$ between 80% and 86% top-1 accuracy, while 3-model cascades achieve $4.4\times$ above 87% accuracy. We confirm wider applicability and effectiveness of the method on the GLUE benchmark. We release the code to reproduce our experiments in the supplementary material and use only publicly available pretrained models and datasets.

## 1 Introduction

The trade-off between accuracy and efficiency is fundamentally important to deep learning. While state-of-the-art results are achieved by models of ever-growing size, practical applications are constrained by the cost of inference. Reducing the energy consumption is critical for energy-limited systems like wearable and mobile devices. For inference at large scale such as in data centers, minimizing the compute resource requirements is important economically and environmentally, and enables wider adoption of beneficial new technology.

Early exit model cascades are a cheap, simple, and effective approach to improve the accuracy-efficiency trade-off. The cost of training and inference is dominated by the largest model in the cascade, which means that the cost added by smaller models is usually negligible. Pretrained models can be used as is or finetuned. When the input is processed by at least 2 models in the cascade, the predictions can be ensembled. Gontijo-Lopes et al. (2022) show that ensembles are more accurate when the models are more diverse due to the lower error correlation, which can also be leveraged by cascades. Cascading is a complementary approach and

can be combined with many other approaches for increasing efficiency, particularly methods that improve the individual models, such as better architecture and training or model compression. Importantly, model cascades can be applied at any scale and tend to be more effective for larger models as can be seen from our experiments.

Although early exit model cascades are effective, little prior work on them has been published. In Section 2 we identify a lack of comparison between different cascading methods, knowledge gaps about when cascades perform well, and limited evaluation. Furthermore, different works use different datasets and metrics. Efficacy is commonly measured by comparison with models trained for the experiments. Code and models are usually not released, and specifics about model size are not always provided. This makes a comprehensive comparison between different works infeasible.

We address these points by conducting a comprehensive evaluation and creating a solid foundation for future work. The choice of baseline is critical to the empirical evaluation as it determines the measured efficacy. To ensure valid and relevant results, we compare with Pareto fronts of state-of-the-art pretrained models, which represent the best achievable trade-off between accuracy and efficiency. Multiply-Accumulate (MAC) operations are a widely used measure of computation in deep learning because the computation in models is typically dominated by matrix multiplications, which consist entirely of MAC operations. Furthermore, MAC count enables hardware agnostic benchmarking and gives an indication of energy consumption and to a lesser extent inference speed. We therefore focus on MAC count as measure of efficiency in our evaluation. No training is done in our experiments. Instead, we use publicly available datasets and pretrained models. We release our code to make our work easily reproducible. The code is designed so that the results can be updated with new state-of-the-art models.

Most of our experiments are conducted on ImageNet (Russakovsky et al., 2015) due to its significance and the large amount of pretrained models available from PyTorch Image Models (Wightman, 2019). We explore how to build cascades effectively and compare many different cascading methods across continuous Pareto fronts in multiple settings using a diverse selection of models for a thorough and reliable evaluation. We then test their wider applicability on text classification tasks from the GLUE (Wang et al., 2019) benchmark using models from Hugging Face (Wolf et al., 2020). Our contributions are:

- We demonstrate that already 2-model cascades dominate the ImageNet Pareto front.

- We provide insight into how to construct model cascades by investigating how to choose models to combine, what confidence metric to use and whether to aggregate predictions.

- We investigate the impact of a distribution shift on cascades.

- We provide an easily reproducible baseline for future research.

## 2 Related Work

**Accuracy-efficiency trade-off**  Improving the accuracy-efficiency trade-off is a central objective for a wide breadth of research with many different approaches. MobileNetV3 (Howard et al., 2019) and EfficientNet (Tan & Le, 2019) represent how architectures have become highly optimized. Model efficiency can be improved further by applying compression techniques like quantization and pruning (Han et al., 2016). Once-for-all (Cai et al., 2020) utilizes progressive shrinking of input resolution, kernel size, network width and depth together with knowledge distillation to obtain more efficient models than conventional neural architecture search. Training is very important with recent advances most notably through data augmentation (Shorten & Khoshgoftaar, 2019) and pretraining on more data (Ridnik et al., 2021), which is enabled further by self-supervised (Devlin et al., 2019) and semi-supervised (Pham et al., 2021) methods. Many other procedures exist such as model soups (Wortsman et al., 2022), which fine-tunes a model with multiple hyperparameter configurations and averages the weights. We focus on early exiting from a cascade of models, which is a complementary approach that is comparatively simple yet effective.

**Dynamic models**   More closely related to our work are dynamic neural networks (Han et al., 2021; Xu & McAuley, 2023). An example of dynamic depth is SkipNet (Wang et al., 2018b), which adds gating modules and a learned skipping policy to skip network layers based on the input. BranchyNet (Teerapittayanon et al., 2016) adds branches to the original net for early evaluation and exits when prediction entropy is below a threshold. Similarly, Shallow-Deep Networks (Kaya et al., 2019) insert early classifiers. Wołczyk et al. (2021) improve early exiting from within a model by recycling predictions of earlier classifiers. However, Huang et al. (2018) describe how early classifiers lack coarse features, and training the model for early classification can lower the accuracy of later classifiers. Furthermore, modern convolutional neural networks are highly optimized and their depth, width, and resolution are carefully scaled, gradually lowering the resolution and increasing the number of channels as features become more complex (Tan & Le, 2019). For early classification outputs, this optimization and scaling is disrupted, which results in a lower accuracy than a separate model at equal compute effort can achieve. We demonstrate experimentally that a smaller accuracy difference between classifiers allows for more frequent early exiting. This motivates using a separate model for early classification, which enables us to increase the early exit rate at the cost of a small overhead when forwarding to the larger model since features are no longer shared between classifiers. However, early exiting from within a single model and between cascaded models are two separate and complementary approaches that can be combined (Bolukbasi et al., 2017).

**Early exit model cascades**   Cascades are commonly used in machine learning and have been popularized by influential works such as Viola & Jones (2001). We focus on a specific type of cascade that uses multiple models in sequence with the capability to exit early if confident enough in a prediction. Park et al. (2015) propose the big/LITTLE architecture, which is a cascade of a small and large model with static or dynamic softmax margin threshold as the exit condition to reduce energy consumption. Wang et al. (2018a) demonstrate reduced computational cost by cascading up to 3 models. They compare different classifiers for the objective of achieving the desired accuracy, which includes a trained decision network, cost-oblivious grid search for both entropy and maximum softmax thresholds, and cost-aware learned entropy threshold. They find that only the last three methods manage to satisfy the accuracy objective, that entropy is a better confidence metric than maximum softmax, and that the cost-aware method performs better. Guan et al. (2018) train an agent to exit between models to reduce the computational cost. Similarly, Bolukbasi et al. (2017) present trained policies to exit early between as well as within models for a trade-off between accuracy and inference time. Streeter (2018) proposes an algorithm to automatically construct efficient cascades from a selection of available models. Inoue (2019) shows that exiting early from an ensemble of 10 or 20 models can greatly reduce the computational cost while preserving most of the static ensemble's accuracy. Inoue averages the softmax to ensemble the predictions and exit based on a confidence interval condition. Wang et al. (2022) cascade up to 4 models of the same architecture families such as EfficientNet, ensemble by averaging the logits, use maximum softmax as confidence metric, and optimize for accuracy and computational cost with exhaustive search. They also demonstrate the efficacy of self-cascading by varying the input resolution for the same model.

From prior work, we notice a lack of comparison between different cascading methods, knowledge gaps about when cascades perform well, and limited evaluation. We address this by comparing many different cascading methods in multiple settings using a diverse selection of pretrained models for a thorough and reliable evaluation. We investigate when cascades work well and unlike prior work we evaluate improvement across continuous Pareto fronts. No training is done in our work. Instead, we use publicly available state-of-the-art models and release the code to reproduce our results, thereby providing a strong baseline that is currently missing for future related research.

## 3   Method

The idea behind early exit cascades is that inference can be done on a smaller model first and if confident in its prediction we can save time and computation by exiting early and skipping the larger model. A cascade contains at least 2 pretrained models. A condition is needed to decide whether to exit early. This condition uses a confidence score computed from the model prediction and a threshold set in advance that needs to be satisfied. Algorithm 1 shows the implementation of the cascading method we found to be most effective.

---

**Algorithm 1** Early exit model cascade with maximum softmax confidence metric and no ensembling

---

**Require:** input tensor $\mathbf{X}$, models $\{M_1, ..., M_n\}$ ordered by increasing cost, thresholds $\{t_1, ..., t_{n-1}\}$, $n \geq 2$

    **for** $i = 1, ..., n$ **do**

        $\boldsymbol{z}_i = M_i(\mathbf{X})$

        $\boldsymbol{p}_i = \mathrm{softmax}(\boldsymbol{z}_i)$

        **if** $i == n$ **or** $\max(\boldsymbol{p}_i) \geq t_i$ **then**

            return $\arg\max(\boldsymbol{p}_i)$                         ▷ cascade returns predicted class

---

**Cost**    Let $C_i$ be the cost of a model $M_i$ which represents metrics we want to improve, such as the number of Multiply-Accumulate (MAC) operations or time of inference. We order the models within a cascade so that $C_i < C_{i+1}$, meaning the cheaper model makes a prediction first. When there are $n$ models in the cascade, there are $n$ possible costs when inferring a single input. In the case $n = 2$, when an early exit is made, the cost is $C_1$, otherwise, the cost is $C_1 + C_2$. They represent the lower and upper bound for the cascade. Assuming we exit early after the first model at a rate $\varepsilon_1$, the cost becomes $C_1 + (1 - \varepsilon_1)C_2$ on average. We want to evaluate how efficient cascades are by comparing them to a baseline at equal accuracy. The improvement factor $I$ by which we reduce the cost is $I = C_b/(C_1 + (1 - \varepsilon_1)C_2)$, where $C_b$ is the baseline cost and $\varepsilon_1$ is set so that the cascade matches the baseline accuracy. Therefore, the improvement we can achieve is limited by the early exit rate we can reach, and the cost difference between the models, where larger is better for both.

**Early exit condition**    We need a condition to decide when to exit early. Such a condition is typically evaluated by obtaining a confidence score from the model prediction and comparing it to a threshold $t$. This is done after each model in the cascade except the last, resulting in $n-1$ thresholds. For classification tasks, the model output for an input tensor $\mathbf{X}$ is usually a logits vector $\boldsymbol{z}$ and the predicted class is $\arg\max(\boldsymbol{z})$. We can use the softmax function

$$\mathrm{softmax}(z_i) = \frac{e^{z_i}}{\sum_{j=1}^{K} e^{z_j}} = p_i \tag{1}$$

to transform the logits into pseudo-probabilities $\boldsymbol{p}$ for all $K$ classes. One confidence metric is the maximum softmax, which we can use as the probability that the prediction is correct. We exit early if $\max(\boldsymbol{p}) \geq t$. Various other conditions exist in related literature (Han et al., 2021; Xu & McAuley, 2023) and the entropy threshold appears to be the most common. There, after applying the softmax function, we can compute the Shannon entropy

$$H(\boldsymbol{p}) = -\sum_{i=1}^{K} p_i \log(p_i). \tag{2}$$

A lower entropy represents higher confidence in the prediction and we exit early when $H(\boldsymbol{p}) \leq t$. Other confidence metrics are the logits margin (Streeter, 2018) and the softmax margin (Park et al., 2015), which are the largest logit or softmax minus the second largest respectively. They represent how much more confident the model is in its first prediction and we exit if the margin is above or equal to a threshold.

**Ensemble**    When the cascade does not exit after the first model, we have multiple predictions that we can ensemble in the hope of improving performance. Typical ways to ensemble models are plurality vote (Hansen & Salamon, 1990) or averaging of the outputs (Bishop et al., 1995). To mean ensemble the outputs, we compute the arithmetic mean of softmax or logits. An intuitive alternative is to compare the prediction confidence of all models and choose the most confident, which we call comparison ensemble. Weighting or calibration can be used when model confidences are mismatched so that the most overconfident model does not dominate the ensemble. Guo et al. (2017) demonstrate that temperature scaling is an effective calibration tool and Ashukha et al. (2020) recommend it for ensembles. Temperature scaling is achieved by dividing the logits by a temperature factor $T$ so that the calibrated probabilities become

$$\hat{p}_i = \frac{e^{\frac{z_i}{T}}}{\sum_{j=1}^{K} e^{\frac{z_j}{T}}} \tag{3}$$

where $T$ is set to minimize the negative log-likelihood on a validation set. We examine the calibration of models used in our ImageNet experiments in Appendix E.

**Evaluation**  The performance of a cascade can be evaluated on a validation set. First, the logits are obtained for each model with a batched forward pass on the entire validation set. From the logits, the confidence score of each input can be calculated according to the used metric. For 2-model cascades, a $k \times 3$ array is constructed containing the confidence score for the first model and prediction correctness for both models for each of the $k$ examples in the validation set. The array is sorted along the $k$ dimension according to confidence score from worst to best. To obtain the cascade accuracy, the cumulative sum of correct examples from the second model and the reversed cumulative sum for the first model are added together and divided by $k$. The cumulative sums represent the number of correct predictions for each model. Now, both the threshold boundaries and cascade accuracy are known for all $k + 1$ possible early exit rates. The average inference cost can be obtained with linear interpolation from $C_1$ at 100% early exit to $C_1 + C_2$ at 0% early exit.

The time complexity for all operations is $O(k)$ except for the sort, which is $O(k \log k)$. Therefore, evaluation is very cheap for 2-model cascades. The process is dominated by the model inference on the validation set. However, the complexity increases exponentially as more models are added to the cascade and becomes $O(k^{n-1} \log k)$ for $n$ models with $n-1$ thresholds. Evaluation with more models can be done by sweeping the thresholds from the last to the first. Heuristic search can be used to reduce the complexity. Alternatively, in order to avoid the exponentially increasing complexity altogether, all thresholds can be set to be the same. However, this would negatively impact performance. When ensembling is used, this basic recipe needs to be adjusted by using the ensemble prediction correctness and confidence score instead.

We evaluate Pareto improvement by comparing the cost of a cascade to achieve the same accuracy as a baseline Pareto front of single models. We conduct Pareto optimal linear interpolation between base models to enable continuous comparison instead of just at specific models. This linear interpolation represents the trade-off achieved when randomly selecting one of the two models with varying probabilities.

**Threshold selection**  A fixed threshold can be selected for testing and deployment based on the evaluation. This is done once ahead of time. There is a constant overhead during deployment to compute the confidence score and compare it to the threshold, which is extremely small compared to the cost of the models. The selection process depends on the desired trade-off between efficiency and accuracy. For example, one may select the threshold which achieves the greatest Pareto improvement, or the greatest reduction in inference cost while preserving the accuracy of the best model. One concern of a fixed threshold is that the validation data may not align with the real world data encountered during deployment. To address this, we analyze the impact of distribution shifts at the end of Section 4.1. Alternatively, dynamic thresholds can be used, for example by defining a computational budget that must be satisfied (Huang et al., 2018).

## 4  Experiments

In order to evaluate the impact of early exit model cascades, we verify whether they reliably outperform state-of-the-art single models across entire Pareto fronts on different tasks. We rigorously compare many different cascading methods to determine the most effective. To understand when cascades work well, we figure out how properties such as the size and accuracy of cascaded models affect performance. We begin by conducting a set of experiments on ImageNet (Russakovsky et al., 2015) (Section 4.1) due to its relevance and evaluate robustness to distribution shifts on ImageNetV2 (Recht et al., 2019). As a step to indicate the general utility of model cascades, we demonstrate Pareto improvement on text classification tasks from the GLUE (Wang et al., 2019) benchmark (Section 4.2). All pretrained models used for image and text classification are publicly available from PyTorch Image Models (Wightman, 2019) and Hugging Face (Wolf et al., 2020) respectively. The code to reproduce these experiments is available in the supplementary material.

### 4.1 Image classification on ImageNet

We select from 668 pretrained models with tabulated accuracy for the ImageNet validation set from PyTorch Image Models (Wightman, 2019). For the computational cost of models, we obtain MAC count with the *fvcore* library. To acquire the time cost we use benchmark numbers in inferred samples per second on an RTX 3090 with NHWC data format and automatic mixed precision from Wightman (2019). From this, we can establish baseline Pareto fronts for both MAC and time cost. We infer the ImageNet validation set for models at the Pareto fronts to confirm their accuracy and obtain logits, from which we compute the prediction correctness and confidence score for every image. From these, we can efficiently compute exact accuracy, threshold, and average inference cost at all possible early exit rates for every cascade. Threshold selection can then be made based on the desired trade-off. For example, we choose the threshold at the point of maximal validation Pareto improvement for our ImageNet test server submissions.

As a strong and continuous baseline, we use the Pareto front established by linear interpolation between pretrained model pairs, which represents the trade-off achieved when randomly selecting one of the two models with varying probabilities. No interpolation is done for cascades in order to have a direct comparison between individual cascades and random model selection. We evaluate improvement by cascade Pareto fronts as a factor by which average inference cost in MAC or time is reduced to achieve the same accuracy. The average cost represents the total cost for the entire validation set at specific thresholds divided by the number of images.

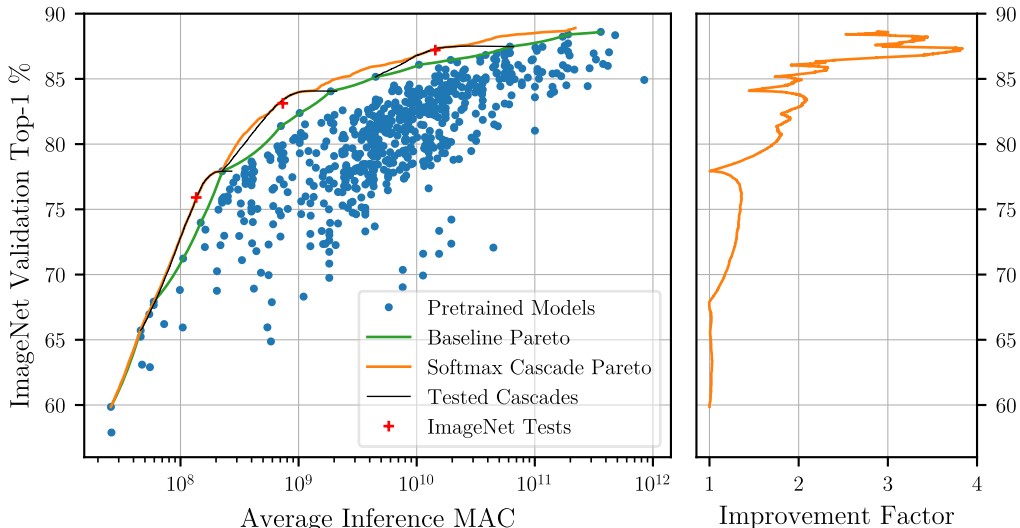

Figure 1: The maximum softmax confidence metric without calibration and ensembling is one of the simplest yet most effective for cascading and achieves Pareto improvement across most of the MAC range with 2-model cascades. The pretrained models show that the accuracy benefit from increasing model size diminishes. Cascade performance improves as the accuracy difference between models decreases compared to the MAC difference. Three points at the cascade Pareto front with locally maximal improvement were evaluated on the ImageNet test set. The results are summarized in Table 1 and confirm the validation results. Notable are that cascading also increases the top accuracy by 0.306% to 88.90%, and that the improvement factor drops to almost 1 at 77.92% accuracy due to an outlier model, a MobileNetV3-Large with much higher accuracy than similarly sized models, which was achieved by pretraining on ImageNet-22K (Ridnik et al., 2021).

For the trade-off between accuracy and MAC count, the baseline Pareto front consists of 27 models. For 2-model cascades, there are $\binom{27}{2} = 351$ possible combinations. Figure 1 shows the Pareto front and improvement factor for the cascading method which uses maximum softmax confidence without calibration or ensembling. The cascades achieve improvement along most of the Pareto front with a reduction in computational cost by a factor of up to 3.83 for the same accuracy. This is achieved by the cascade containing the models EfficientNet B4 NS (Tan & Le, 2019), a convolutional neural network pretrained on 300M unlabeled

Table 1: Three points at the locally maximal improvement from the softmax 2-model cascade Pareto front were evaluated on the ImageNet test set as shown in Figure 1. They were chosen as the strongest points at the cascade Pareto front to test against validation set overfitting. Each point corresponds to a cascade with a specific threshold. Values are rounded to 4 significant digits.

| Cascade Models | | Validation | | Test | | Softmax |
|---|---|---|---|---|---|---|
| First | Second | Top-1 ↑ | MAC ↓ | Top-1 ↑ | MAC ↓ | Threshold |
| MN3-S 0.75 (Howard et al., 2019) | MN3-L (Ridnik et al., 2021) | 75.88% | 136.2M | **75.91%** | **135.9M** | 0.4551 |
| MN3-L (Ridnik et al., 2021) | EN B3 NS (Xie et al., 2020) | **83.39%** | **732.4M** | 83.13% | 735.2M | 0.7112 |
| EN B4 NS (Xie et al., 2020) | BEiT$_{224}$-L (Bao et al., 2022) | **87.32%** | 14.39G | 87.21% | **14.38G** | 0.5658 |

images from the JFT dataset (Hinton et al., 2015; Chollet, 2017), and BEiT$_{224}$-L (Bao et al., 2022), a vision transformer pretrained on ImageNet-22K, which shows the benefit of model diversity. Model families which contribute to the cascade Pareto front in descending order of importance are EfficientNet NoisyStudent (Xie et al., 2020), MobileNetV3 (Howard et al., 2019; Ridnik et al., 2021), BEiT (Bao et al., 2022), TinyNet (Han et al., 2020b), GhostNet (Han et al., 2020a) and LeViT (Graham et al., 2021).

To test against validation set overfitting, points of the cascade Pareto front with locally maximal improvement were evaluated on the ImageNet test set. As shown in Table 1, the test set performance is close enough to confirm the validation set results. However, test accuracy is slightly lower on average. Possible explanations are that the optimal points are shifted for the test set, that the cascade Pareto front envelops noise, and that the pretrained models themselves are slightly overfitted to the validation set. To investigate the last point, the pretrained EfficientNet B4 NoisyStudent was evaluated on the ImageNet test set and achieved a top-1 accuracy 0.04% lower than on the validation set, which accounts for almost half of the 0.09% difference for the 3rd test point.

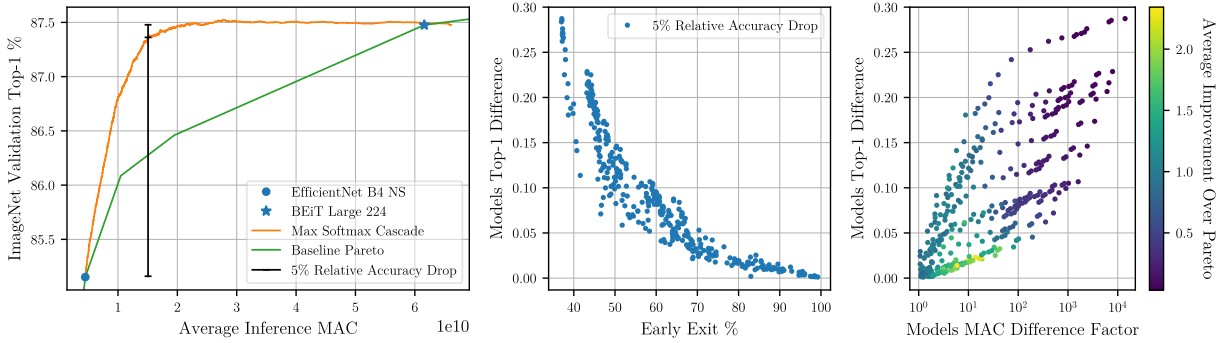

Figure 2: **Left:** The cascades tend to maintain the accuracy of the better model for a plateau or even exceed it before accuracy starts to break down. **Middle:** Each point represents the largest early exit rate before cascade accuracy drops by 5% of the top-1 difference below the better model. **Right:** The average Pareto improvement of all individual cascades. While a lower accuracy difference between cascaded models is always desirable, there exists a sweet spot for the size difference. While the length of the plateau before accuracy drops increases with smaller accuracy difference between models, for the baseline Pareto this plateau grows with the size range as the accuracy benefit of increased model size diminishes.

Figure 2 shows the relationship between the accuracy difference of the models in the cascade and the early exit rates achievable by cascades before accuracy starts to break down. The cascades perform better when the accuracy difference between models is small while the MAC difference is large. Therefore, the Pareto improvement is minimal in the lowest MAC regime and becomes larger at higher MAC regimes because of the diminishing accuracy benefit from increased model size, which can be observed in Figure 1.

**Cascading method comparison**  We compare the 4 confidence metrics entropy, maximum softmax, softmax margin, and logits margin both without ensembling and with comparison ensembling enabled by calibration using temperature scaling. Furthermore, using the maximum softmax confidence metric we evaluate mean ensembling for both logits and softmax, with and without temperature scaling. This results in 12 different cascading methods compared in Figure 3.

From the methods without ensembling, the entropy confidence metric performs worst. Entropy is influenced by probabilities assigned to false labels while we are only interested in whether the predicted label is correct for early exiting. Softmax margin is the only cascading method that achieves an improvement >1 across the entire Pareto front—except at the smallest model, whose computational cost cannot be reduced further with cascading. It also reaches the largest average Pareto improvement of all 12 methods with 1.502× across the entire accuracy range. This is because softmax margin dominates below 78% accuracy, which represents almost 2/3 of the accuracy range but only about 1/4 of the log-scaled MAC range. The maximum softmax metric has the second-largest average improvement at 1.497× but shows the most MAC range dominance of all methods.

When ensembling, calibration may be required, because otherwise there is no guarantee that the uncalibrated confidence scores of the different models are comparable. Temperature scaling was done on the validation set, which means that the results for methods using temperature scaling represent an ideal scenario where the calibration is optimally fitted to the data. Yet, calibration proves ineffective for mean ensemble cascades and the comparison ensemble methods achieve the least improvement of all. Mean ensembling shows potential at the upper end of the accuracy range. Therefore, we use the maximum softmax confidence metric, which is worse overall but better at higher accuracies than the softmax margin (see Appendix C for details). While the mean softmax method performs better overall, uncalibrated mean logits ensembling achieves the best average improvement above 86% accuracy of all cascading methods, as well as the best top accuracy of 89.03%. A possible explanation for the poor performance of ensembling is that Pareto optimal cascade settings rely on high early exit rates, where only the least certain predictions are forwarded. These are less useful for or may even hurt ensembling.

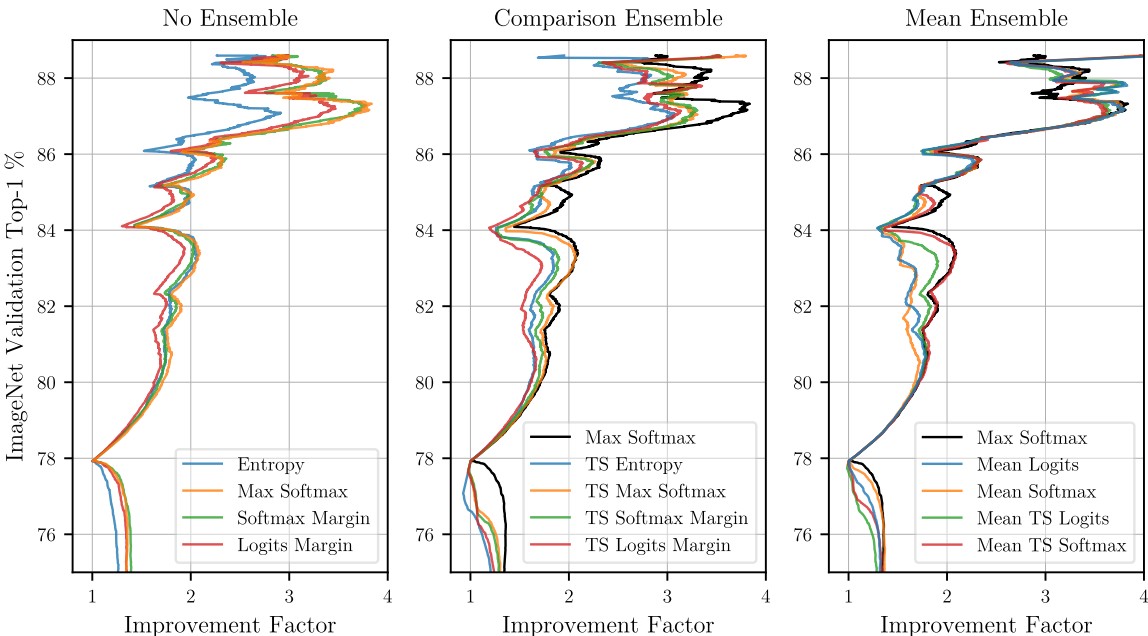

Figure 3: A comparison of Pareto improvement achieved for the 12 tested cascading methods. To improve legibility, only the accuracy range above 75% is shown as improvement achieved below that point is small for all methods. The maximum softmax method from the left is plotted again as a black line to have a baseline for comparison. TS stands for temperature-scaled.

**3-model cascades**   Cascading more models increases the complexity since well-suited models are required for improvement and multiple thresholds need to be selected, which increases the problem dimensionality. The ImageNet validation set consists of 50000 images, which means that there are 50001 choices per threshold ranging from always to never exiting early. For the 27 Pareto optimal models, there are $\binom{27}{3} = 2925$ possible combinations. Since computing cascade performance is cheap, an exhaustive search is still feasible for 3-model cascades and has the advantages of simplicity and a guarantee that the optimum will be found. To do so, for every model combination we conduct a sweep on the first threshold. We can speed up the process by only sampling every tenth sorted value for the first threshold since neighboring values are almost identical.

Interestingly, ensembling performs slightly worse relative to no ensembling for 3 instead of 2 models, and mean softmax is now superior to mean logits. The maximum softmax method achieves the largest average Pareto improvement of all cascading methods both across the entire accuracy range and above 80% only. Figure 4 shows that there is diminished benefit from adding a third model but clear improvement at higher accuracies. 2-model cascades offer a strong and practical baseline, but for optimal performance adding more models should be considered while keeping in mind the trade-off between diminishing returns and increased complexity, which could also harm generalizability. To test this, we evaluated the point of the 3-model maximum softmax Pareto front with the largest improvement on the ImageNet test set. This is the cascade with models EfficientNet B4 NS (Tan & Le, 2019), BEiT$_{224}$-L (Bao et al., 2022), and EfficientNet L2 NS at the thresholds $t_1 = 0.6299$ and $t_2 = 0.4905$. While test accuracy at 87.86% is only 0.08% lower than on the validation set, the average MAC count is also 1.1% larger at 26.32G. This indicates a slightly worse generalization than the 3rd test result in Table 1 for a cascade with only the first 2 models.

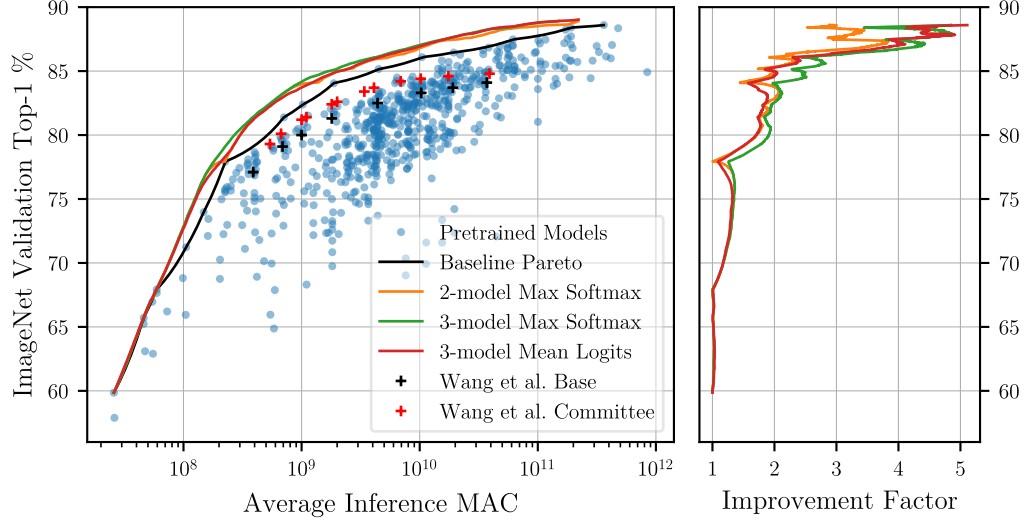

Figure 4: 3-model cascades start to outperform 2-model cascades at higher MAC regimes. **Left:** We compare our results to Wang et al. (2022), who use the mean logits ensemble method. **Right:** The maximum softmax method achieves the best performance of all cascading methods and outperforms the mean logits method used by Wang et al. (2022) across most of the Pareto front.

Wołczyk et al. (2021) report results for exiting early from within a model on ImageNet. They compare three different methods for adding multiple early classifiers to a pretrained ResNet-50 (He et al., 2016): Shallow-Deep Networks (Kaya et al., 2019), Patience-based Early Exit (Zhou et al., 2020), and Zero Time Waste (Wołczyk et al., 2021). For an average reduction in inference MAC by 25%, the accuracy drops by 6.1%, 10.4%, and 5.7% for the respective methods. Accuracy degrades more rapidly as the average inference cost is reduced further. The inference cost of a ResNet-50 is 4.111 GMAC. At this point of our baseline Pareto front we obtain a 59% reduction in inference cost while achieving equal accuracy for 3-model maximum softmax cascades.

**Inference time** Radosavovic et al. (2020) show that the number of MAC operations is an imperfect predictor for GPU inference speed. Therefore, we also evaluate cascade performance for the trade-off between accuracy and speed measured in average time per inference. Figure 5 shows that 2-model cascades already achieve improvement everywhere except near the outlying LeViT-128S (Graham et al., 2021). As can be seen in greater detail in Appendix A, maximum softmax again performs best of all tested cascading methods, which demonstrates its reliability and consistency. Notably, it outperforms mean ensemble methods even above 86% accuracy for inference speed, achieving an average improvement of 2.618× compared to 2.203× by the mean logits cascading method. The baseline Pareto front consists of 25 models. There is a large model overlap between MAC and speed Pareto fronts. The main difference is that EfficientNet (Tan & Le, 2019; Xie et al., 2020) and the related TinyNet (Han et al., 2020b) architecture families perform worse for inference speed while LeViT (Graham et al., 2021) and ConvNeXt (Liu et al., 2022) perform better.

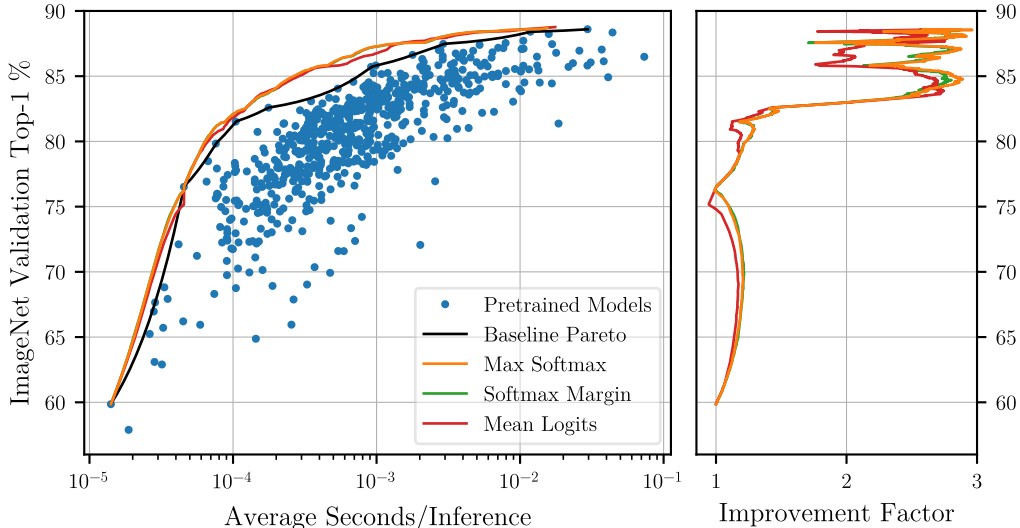

Figure 5: Inference speed is measured in average time per image on an RTX 3090 GPU. The images are batched with a batch size of 1024 or smaller when necessary to fit the GPU memory. Therefore, the actual latency is larger. Less Pareto improvement is achieved than for MAC count because the time cost Pareto front spans 3 orders of magnitude instead of 4. The 2-model cascades perform particularly well at concave points of the baseline Pareto front.

**Robustness to distribution shifts** Training data may not align with real-world inputs. A concern is that model cascades are vulnerable to distribution shifts, particularly when the data used to determine the threshold differs from the deployment data. We test robustness by evaluating Pareto improvement on the ImageNetV2 MatchedFrequency dataset (Recht et al., 2019) for identical cascades and threshold ranges that make up the original ImageNet cascade Pareto front for the maximum softmax confidence metric. Figure 6 shows that while the distribution shift harms cascade performance, they remain beneficial and achieve improvement across most of the Pareto front. Two contributing factors that decrease cascade performance are that model accuracies on ImageNetV2 are $7.2 - 13.8\%$ lower and accuracy differences between the models are larger with the overall accuracy range increasing by 5.7%. Furthermore, model accuracy is no longer monotonically increasing with size. Additional analysis, which shows that the distribution shifted ImageNet cascade Pareto front resembles the new ImageNetV2 cascade Pareto front, can be seen in Appendix D.

## 4.2 Text classification on GLUE benchmark

Early exit model cascades are effective for image classification with both convolutional neural networks and vision transformers. We expect that the cascades also work in other domains where prediction confidence or quality can be quantified. Indeed, beneficial early exiting has been demonstrated for image segmentation (Wang et al., 2022) and reinforcement learning (Wołczyk et al., 2021). As a first step to test that the identical

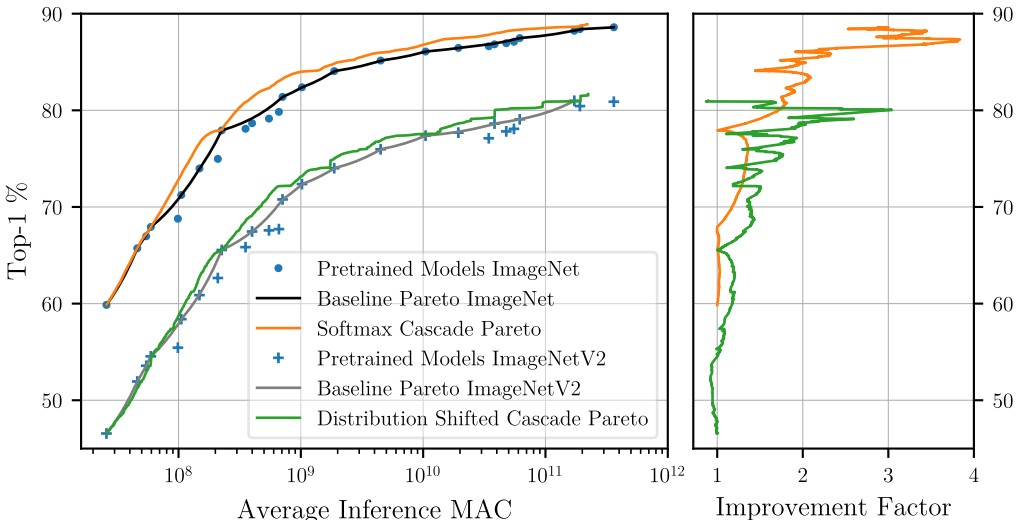

Figure 6: Robustness to distribution shifts is evaluated by comparing the Pareto improvement achieved for identical cascades and threshold ranges on ImageNet and ImageNetV2.

cascade structure can be used in a different domain, we experiment on the text classification tasks SST-2 (Socher et al., 2013), MRPC (Dolan & Brockett, 2005), QNLI (Rajpurkar et al., 2016) and QQP (Chen et al., 2018) from the GLUE (Wang et al., 2019) benchmark. The results are shown in Figure 7 and confirm the efficacy of early exit cascades. Since the tasks are binary classification, the 4 confidence metrics entropy, maximum softmax, softmax margin, and logits margin result in identical order when sorting the confidence scores of all predictions, and the resulting cascades are equivalent. We searched for fine-tuned models on Hugging Face (Wolf et al., 2020). Models used include BERT (Devlin et al., 2019), RoBERTa (Liu et al., 2019), DeBERTa (He et al., 2021), ELECTRA (Clark et al., 2020), ALBERT (Lan et al., 2020), DynaBERT (Hou et al., 2020), DistilBERT (Sanh et al., 2019), M-FAC (Frantar et al., 2021), MiniLM (Wang et al., 2020) and XLNet (Yang et al., 2019).

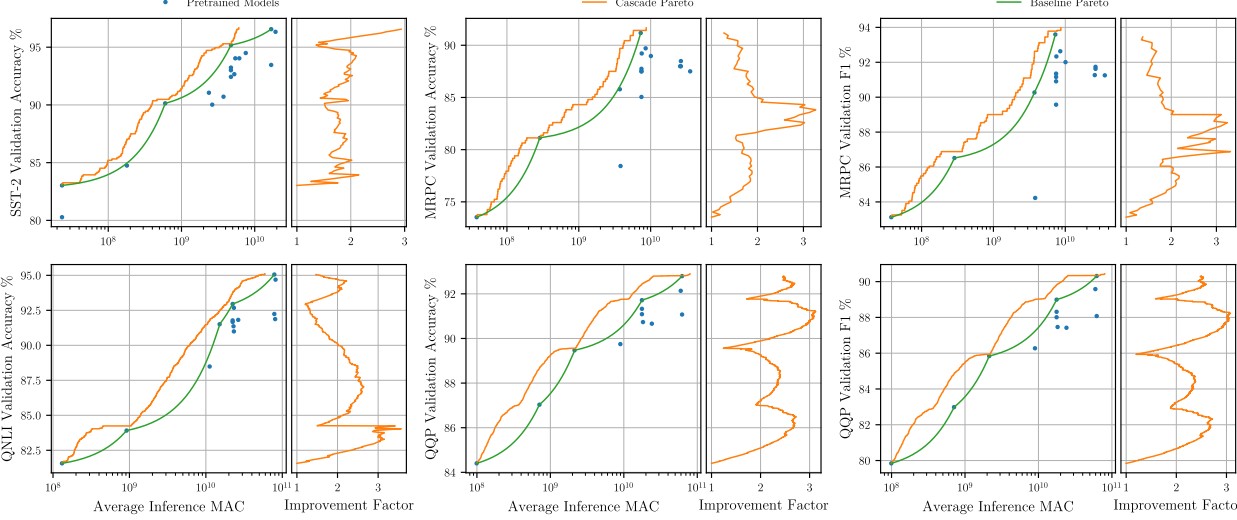

Figure 7: Early exit cascades achieve Pareto improvement for text classification tasks from the GLUE benchmark despite the large accuracy differences between models. The results for the MRPC and QQP tasks are similar for both accuracy and F1 score. 18 models are used for SST-2, 16 for MRPC, 12 for QQP, and 15 for QNLI.

## 5 Conclusion

Early exit model cascades can be used to trade off inference cost against quality, to optimize for maximum Pareto improvement, or to maintain and even exceed the performance of the largest model while lowering cost. We demonstrate the efficacy of early exit cascades on a wide selection of publicly available pretrained models for image classification on ImageNet and text classification tasks from the GLUE benchmark. We cascade Pareto optimal models independent of their architecture or training and provide a strong and reproducible baseline that is currently missing for related research. To investigate the effective design of cascades, we compare various confidence metrics and ensembling strategies. We find that the maximum softmax confidence metric achieves the largest improvement overall while softmax margin excels in low confidence scenarios and the commonly used entropy metric performs worst, that ensembling predictions can increase the maximum accuracy the cascade achieves but fails to improve the Pareto front, and that temperature scaling is ineffective at alleviating this. We demonstrate a relationship between the accuracy difference of cascaded models and the achievable early exit rate as well as the desirable size difference, and show that while distribution shifts degrade cascade performance, cascades remain beneficial.

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

## A    Additional Pareto improvement comparison data

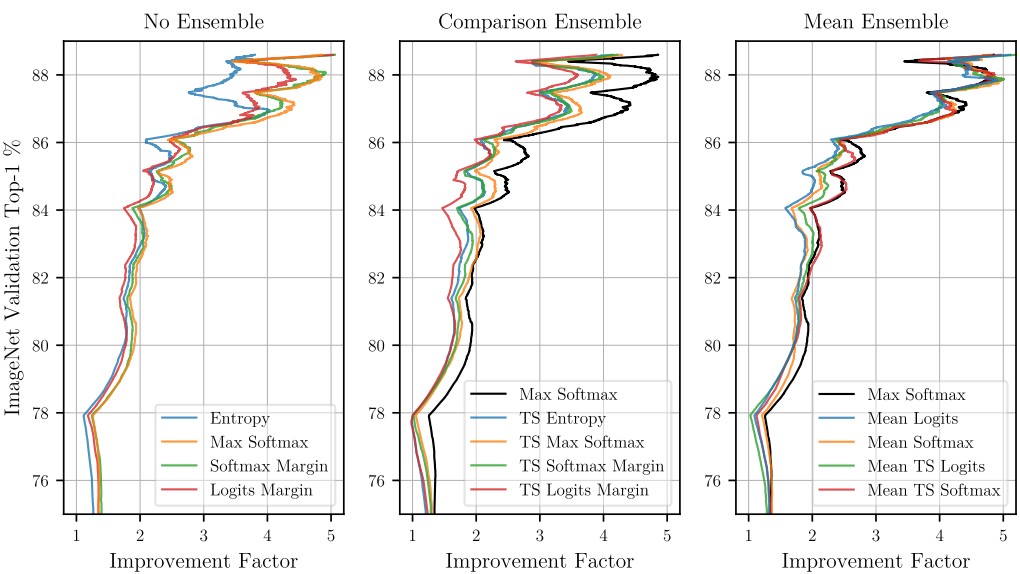

Figure 8: 3-model cascade Pareto improvement for the accuracy-MAC trade-off. Maximum softmax confidence without ensembling is the best cascading method overall and is plotted again in the middle and right as black line to have a baseline for comparison. The softmax margin cascading method performs best in the lower accuracy range and mean ensembling shines at the upper end of the accuracy range.

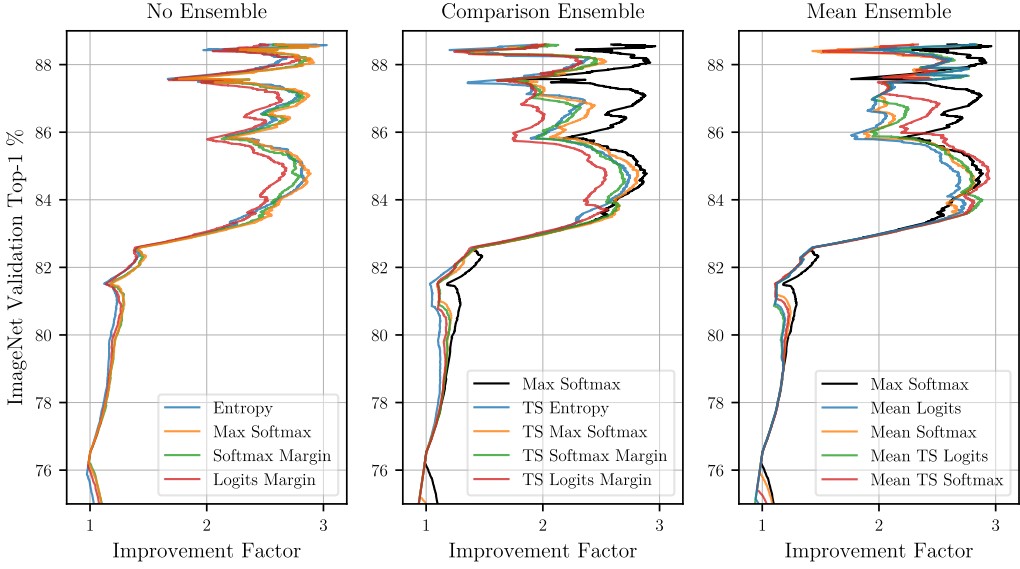

Figure 9: 2-model cascade Pareto improvement for the accuracy-time trade-off. Again, maximum softmax confidence without ensembling is the best cascading method of all and is plotted again in the middle and right as black line to have a baseline for comparison. Notable is that mean ensembling shows worse performance for inference speed relative to the maximum softmax method. Furthermore, the mean softmax method now outperforms the mean logits method. This indicates that cascading methods which use ensembling are less reliable.

# B Top-5 accuracy

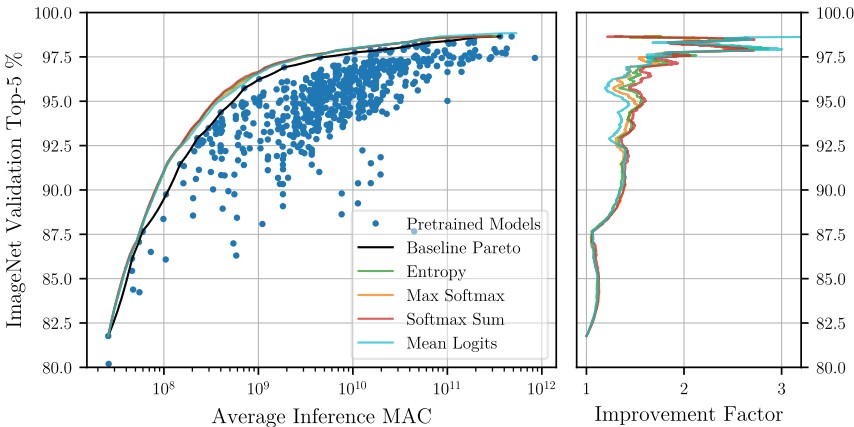

Figure 10: For some applications, such as content or autocomplete recommendation, it is less important that the top-1 prediction is correct and instead it matters more that the correct prediction is among the top. To investigate this scenario, we evaluate the trade-off between top-5 accuracy and MAC count. 2-model cascades perform better at the lower range but worse at the higher range for top-5 accuracy despite the smaller accuracy differences. Here, the entropy confidence metric beats maximum softmax since it takes into account more than the top-1 prediction. However, the top-5 softmax sum, which takes into account all relevant probabilities, outperforms entropy across most of the Pareto front. In an attempt to improve the cascading further, we try mean ensembling. However, neither the mean logits nor the mean softmax (not pictured) method perform well outside of narrow points at the upper limit of the accuracy range.

# C Confidence metric for mean ensembles

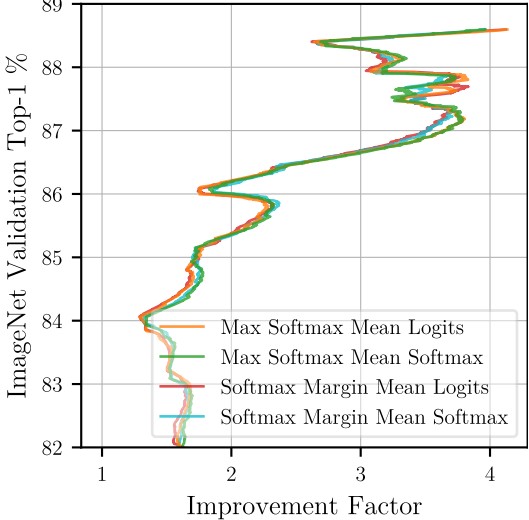

Figure 11: A comparison of confidence metrics for mean ensembled 2-model cascades in the high accuracy regime, where they perform best. While the difference is very small, the softmax margin confidence metric is better overall than the maximum softmax for mean ensembled cascades. However, maximum softmax confidence leads to better improvement in the high accuracy regime, where mean ensembles perform best, and is therefore the metric we selected.

# D Robustness to distribution shifts

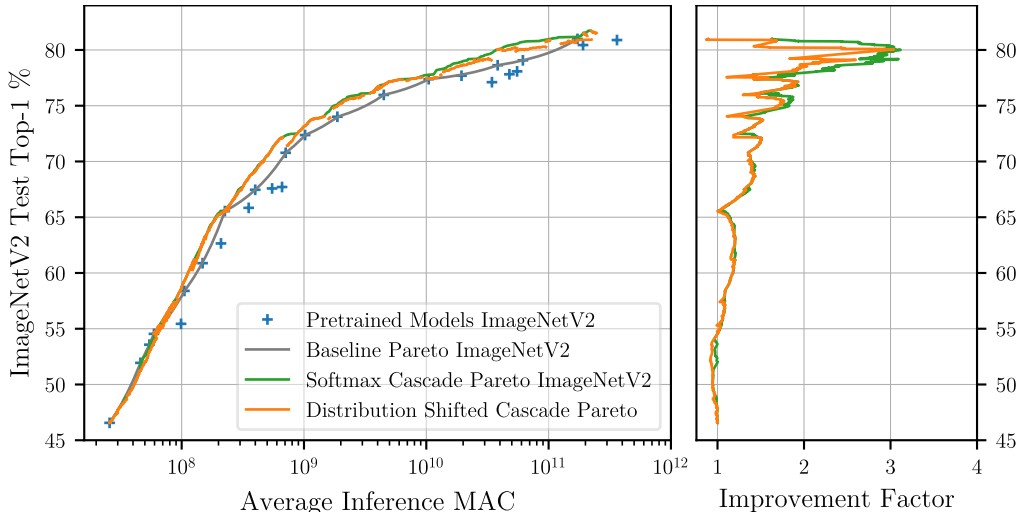

Figure 12: To get a fine-grained view of the impact distribution shifts have on fixed thresholds, we compare the distribution-shifted ImageNet cascade Pareto front with the new ImageNetV2 cascade Pareto front for the maximum softmax confidence metric. The distribution shifted Pareto front is obtained by using the same cascades at identical confidence threshold ranges that make up the original ImageNet Pareto front and evaluating them on ImageNetV2. On the left, we show the individual distribution shifted segments, which represent separate cascades and threshold ranges that make up the original Pareto front. The distribution shift causes these segments to no longer be connected as identical thresholds lead to changed early exit rates for different data and cascades are affected to a different degree. The new ImageNetV2 cascade Pareto front is obtained with the same algorithm that has been used to get the original Pareto front, which obtains the Pareto front across all possible 2-model cascades. The distribution shifted front remains close to Pareto optimal across much of the range.

# E Calibration

Early exit model cascades rely on a relationship between the confidence metric and the model accuracy where more confident predictions should be more accurate. This enables exiting early for predictions more likely to be correct and forwarding predictions less likely to be correct. However, Guo et al. (2017) report that modern networks are poorly calibrated. It is often stated that this miscalibration is typically in the direction of overconfidence (Gawlikowski et al., 2023; Abdar et al., 2021; Wang et al., 2021).

We examine the calibration of models used in our ImageNet experiments in Figure 13 for the maximum softmax confidence metric. The baseline Pareto front consists of 27 models for the trade-off between accuracy and MAC count, and 25 models for the trade-off between accuracy and inference speed. Combined, 43 unique models are cascaded. Their calibration is visualized by comparing moving averages of both accuracy and confidence for 1000 examples across all 50000 examples of the validation set ordered by their confidence.

Contrary to what is often reported, underconfidence is more common than overconfidence among these state-of-the-art models. The largest model families BEiT (Bao et al., 2022), ConvNeXt (Liu et al., 2022), and DeiT3 (Touvron et al., 2022) seem to have better than average calibration and are also among the most modern models used in our experiments.

What matters for cascading is to what extent the miscalibration is a shift and to what extent the positive relationship between accuracy and confidence is disrupted. The former, for example when the model is over- or underconfident, is addressed by shifting the early exit threshold. However, the latter is a concern because it can decrease the model's accuracy for examples that satisfy a threshold. Figure 13 shows that there is a noisy but generally consistent relationship between confidence and accuracy for all models.

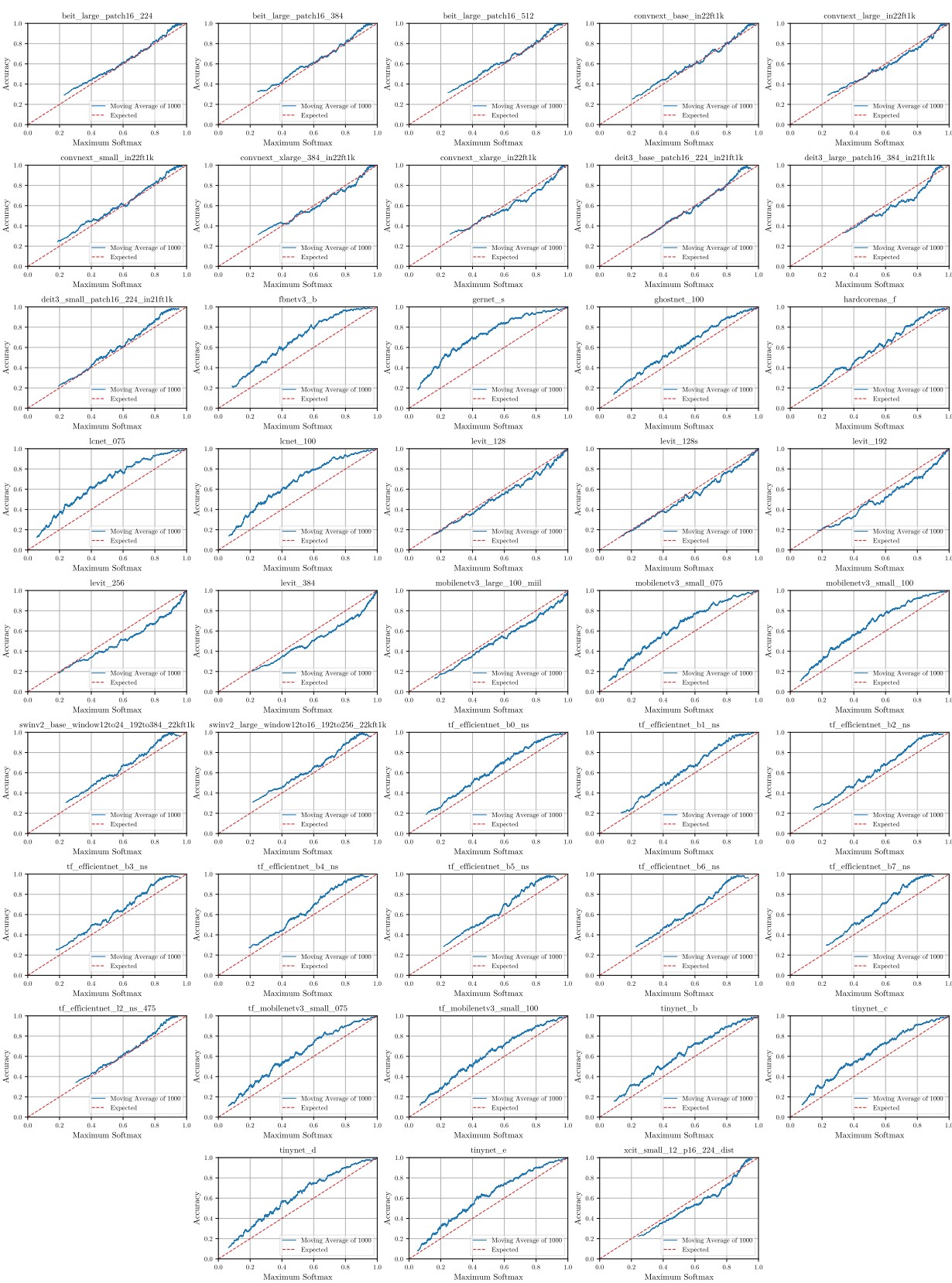

Figure 13: The moving average of 1000 examples for both accuracy and ordered confidence (maximum softmax) for all 43 models used in the ImageNet experiments to visualize how well they are calibrated. The titles are the names of the models according to the PyTorch Image Models library (Wightman, 2019).

We attempt further investigation by inspecting curves of model accuracy above and below respective thresholds representing various early exit rates as shown in Figure 14. For ideal cascade performance we want a confidence score threshold above which every prediction is correct and below which every prediction is false. We compare the uncalibrated with the temperature scaled predictions. Temperature scaling is done on the validation set and therefore maximally fitted to the data, which means this represents an upper limit and calibration performance during deployment is expected to be worse. However, temperature scaled calibration seems to have little impact.

To what extent is calibration desirable? One idea is to simulate perfect calibration by using temperature scaling to set the average maximum softmax equal to the model accuracy and then assume the calibration is perfect by setting the prediction accuracy to be the maximum softmax. Figure 14 shows improved performance for MobileNetV3-Large but worse performance for EfficientNet B4 NoisyStudent. This indicates that calibration is not necessarily desirable. Instead, cascade performance is determined by a combination of the inference cost and accuracy difference between the models, the prediction accuracy above versus below the threshold, and how often the next model makes correct predictions for inputs which do not exit early.

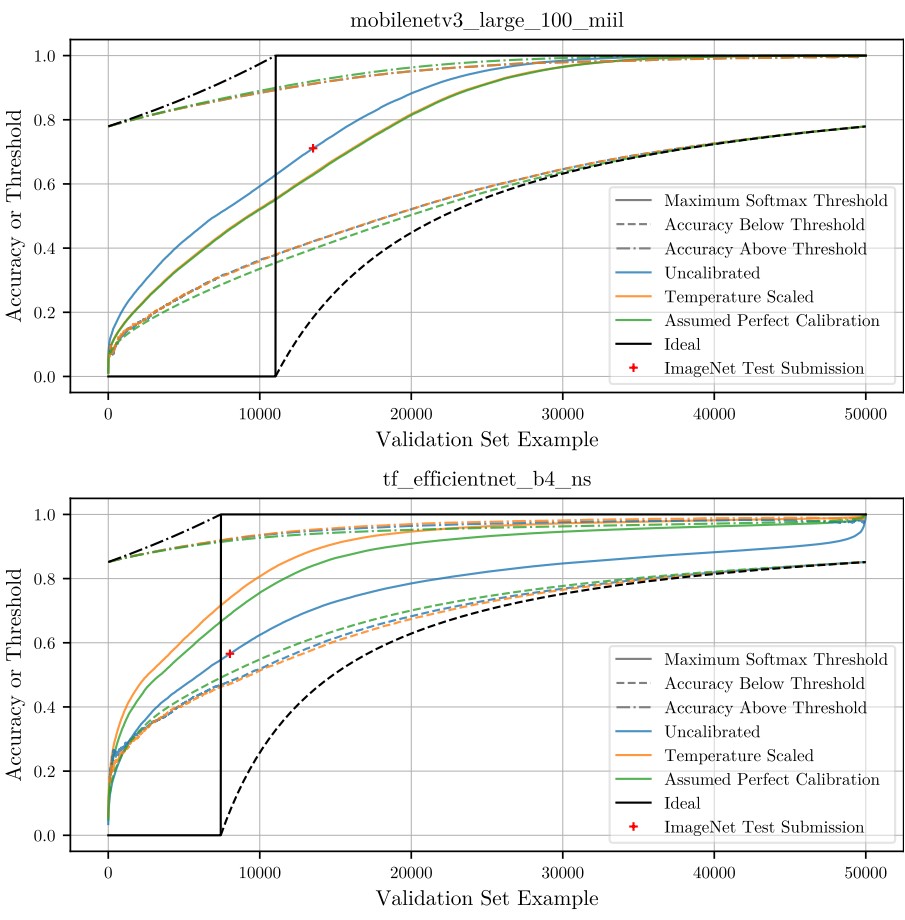

Figure 14: The ordered maximum softmax with prediction accuracy for examples above and below the respective threshold is shown for **uncalibrated** and **temperature scaled** predictions, as well as the **ideal** scenario where there is a threshold above which every prediction is correct and below which every prediction is false, and **assumed perfect calibration** where temperature scaling is used to set the average maximum softmax equal to the accuracy and the prediction confidence is then treated as the prediction accuracy. The confidence threshold used for **ImageNet test submissions** as listed in Table 1 is highlighted to show points of maximal Pareto improvement.

