# OpenReview forum: "Efficient Inference With Model Cascades"
_TMLR — Accepted by TMLR_

### Review · Reviewer_aYFZ · 2023-06-19

**Summary Of Contributions:**

In this paper, the authors propose a simple yet effective method for model cascades. The proposed method combines two or more models, and the inference process terminates early when the Softmax output from a model is above a threshold. The authors conduct experiments with hundreds of models on both image and text datasets to verify the effectiveness of the proposed method, demonstrating good performance on the accuracy-efficiency trade-off. The code for the proposed method will be made public as a reproducible state-of-the-art baseline for future research.

**Audience:**

Yes

**Claims And Evidence:**

Yes

**Requested Changes:**

1. The authors give six contributions in Introduction. It would be beneficial to condense them into three or four key points.
2. Some important citations should be supplemented with additional information.

**Strengths And Weaknesses:**

Strengths:
1. The paper is well-organized and easy to follow.
2. Although the proposed method is simple, it demonstrates its effectiveness in Experiments.
3. The paper verifies some current used techniques and establishes a reproducible baseline, which are both useful for future research.

Weakness:
1. What are "Pareto fronts" and "Multiply-Accumulate (MAC)"? The critical citations are missing.

---

> ### Author Response · Authors · 2023-07-15
>
> Thank you for the feedback! We appreciate that you took the time to review our submission.
>
> > What are "Pareto fronts" and "Multiply-Accumulate (MAC)"?
>
> We have added explanations for both when they are first mentioned in the introduction. Pareto fronts represent the boundary of the best achievable trade-off between conflicting objectives, in our case efficiency and accuracy. If a point is at the Pareto front, no other point surpasses it in both accuracy and efficiency. MAC operations are commonly used in deep learning to measure the computation for a single forward pass of a model because the computation is dominated by matrix multiplications, which consist entirely of MAC operations.
>
> > The authors give six contributions in Introduction. It would be beneficial to condense them into three or four key points.
>
> We have changed our contributions as follows:
> 1) We demonstrate that already 2-model cascades dominate the ImageNet Pareto front.
> 2) We provide insight into how to construct model cascades by investigating how to choose models to combine, what confidence metric to use and whether to aggregate predictions.
> 3) We investigate the impact of a distribution shift on cascades.
> 4) We provide an easily reproducible baseline for future research.
>
> Please let us know if any concerns remain or you have further feedback!

---

### Review · Reviewer_FkTW · 2023-06-20

**Summary Of Contributions:**

This paper validates and explores the design space of model cascades -- combining 2 or more pretrained models with conditional execution to perform standard classification tests. The work thoroughly evaluates the impact on the accuracy-efficiency trade-off, contains abundant ablation study, and provide a solid baseline for the direction of utilizing more than one pretrained models combined with early exits to boost performances. Experiments contain both image classification (ImageNet) and text classification (GLUE) tasks, covering a wide range of usability.

**Audience:**

Yes

**Claims And Evidence:**

Yes

**Requested Changes:**

I think the paper can be much stronger once these issues are addressed and corresponding changes made:

- Clarify how the thresholds are obtained for each experiment
- Address the added cost from finding/optimizing thresholds
- Add additional experiments with fixed thresholds (e.g. 0.5) or fixed exit rate and see how it performs
- Add equation or clear definition of MAC counts

Other, minor issues:
 - In Algorithm 1, it should be made clear that models {M1, ..., Mn} are ordered by model size
 - This sentence is unclear, consider rewrite or further clarify: "No linear interpolation is done between points at cascade Pareto fronts to have a direct comparison between individual cascades and random model selection."
 - Also nuclear what this means: "due to the outlying MobileNetV3-Large pretrained on ImageNet-22K"
 - In 4.2, it's unclear how many models in total are used







**Strengths And Weaknesses:**

## Strengths
 - The paper evaluates a simple idea: model cascades with conditional early exit, and backs it up with abundant experimental results.
 - The goal is clear: demonstrating that model cascades constructed from pretrained models of an existing Pareto front further improves the Pareto front
 - Experiments cover both the image domain and the text domain
 - It further explores ensemble methods and their effects
 - The writing is clear, and the paper is structured to be easy to follow


## Weaknesses
 - The biggest weakness of this paper is that it is unclear how the thresholds are obtained. It seems to me that it is optimized for every cascade and exit strategy, at lest for 2-model cascades that produced Figure 1 and Table 1. That has too problems essentially. Firstly, having to optimize for thresholds reduces the validity of the method — the saved inference compute from early exit would be canceled out from the compute for threshold optimization. Secondly, it is the paper's job to document the process clearly. When said "we choose the threshold at the point of maximal validation Pareto improvement for our ImageNet test server submissions" it remains unclear if the threshold is obtained via a grid search, binary search, or something else. In the 3-model cascade case, it is also unclear how "we set the first threshold"
 - The definition of MAC was never spelled out. Since it is such an important concept throughout the paper, it should be clearly defined early on, with equations and explanations.
 - The conclusion is not very convincing. Although the experiment shows that using cascades can allow for optimizing for maxim Pareto improvement, if it incurs extra compute finding proper thresholds, it will not be worth it eventually.

---

> ### Author Response · Authors · 2023-07-15
>
> Thank you for taking the time to provide thorough feedback, we greatly appreciate it!
>
> > The biggest weakness of this paper is that it is unclear how the thresholds are obtained.
>
> We have added a detailed description at the end of Section 3, which clarifies many of the following points.
>
> > It seems to me that it is optimized for every cascade and exit strategy, at lest for 2-model cascades that produced Figure 1 and Table 1. That has too problems essentially. Firstly, having to optimize for thresholds reduces the validity of the method — the saved inference compute from early exit would be canceled out from the compute for threshold optimization.
>
> > The conclusion is not very convincing. Although the experiment shows that using cascades can allow for optimizing for maxim Pareto improvement, if it incurs extra compute finding proper thresholds, it will not be worth it eventually.
>
> While the threshold should be optimized for the individual cascade to achieve optimal performance, this is only done once offline. The threshold remains fixed during inference. Therefore, threshold selection is not a problem in practice. Models are often trained on billions of examples. The threshold can be selected based on evaluation on a small dataset of say 1000 inputs. If the cascade achieves a 50% reduction in inference cost, it pays for itself during deployment already after 1000/0.5 = 2000 inputs. Often, models are already evaluated on a small dataset to test their accuracy. If this is the case, cascading can be used practically for free to get a major improvement in efficiency. The overhead during deployment - computing the confidence score and comparing it to the threshold - is constant and extremely small compared to the cost of the models. The cost of the threshold selection is dominated by performing a forward pass on the small dataset with the individual models to obtain logits for the evaluation. However, this only needs to be done once per model. The rest of the evaluation can be done by a generic CPU in a fraction of a second for 2-model cascades and gives us the results for all possible choices of the threshold.
>
> > When said "we choose the threshold at the point of maximal validation Pareto improvement for our ImageNet test server submissions" it remains unclear if the threshold is obtained via a grid search, binary search, or something else.
>
> From the evaluation we know the threshold boundaries and cascade accuracy for all possible early exit rates. We select the threshold that results in the largest improvement with a max operation, we do not conduct a search.
>
> > In the 3-model cascade case, it is also unclear how "we set the first threshold"
>
> We have reworded this sentence to express that we conduct a sweep on the first threshold.
>
> > The definition of MAC was never spelled out. Since it is such an important concept throughout the paper, it should be clearly defined early on, with equations and explanations.
>
> We have added an explanation in the introduction when MAC operations are first mentioned.
>
>
> > Clarify how the thresholds are obtained for each experiment
>
> > Address the added cost from finding/optimizing thresholds
>
> > Add additional experiments with fixed thresholds (e.g. 0.5) or fixed exit rate and see how it performs
>
> > Add equation or clear definition of MAC counts
>
> The added explanations should clarify these points. The thresholds are already fixed and we show results for all possible choices.
>
> > In Algorithm 1, it should be made clear that models {M1, ..., Mn} are ordered by model size
>
> We have added this information to the algorithm.
>
> > This sentence is unclear, consider rewrite or further clarify: "No linear interpolation is done between points at cascade Pareto fronts to have a direct comparison between individual cascades and random model selection."
>
> We simplified it as "No interpolation is done for cascades in order to have a direct comparison between individual cascades and random model selection." The cascade Pareto front could be improved further with linear interpolation, which represents random selection between two cascades. However, this is not practical. We only use interpolation for the base models so that the baseline Pareto front becomes continuous to enable comparison everywhere instead of just at specific models.
>
> > Also nuclear what this means: "due to the outlying MobileNetV3-Large pretrained on ImageNet-22K"
>
> We have rewritten this as: "due to an outlier model, a MobileNetV3-Large with much higher accuracy than similarly sized models, which was achieved by pretraining on ImageNet-22K."
>
> > In 4.2, it's unclear how many models in total are used
>
> We have added numbers for each task to the description of Figure 7. The exact models are named in the code in the supplementary material.
>
> Please let us know if anything remains unclear.

---

### Review · Reviewer_RnjH · 2023-07-07

**Summary Of Contributions:**

This paper conducts a fairly thorough investigation of so-called model cascades, summarizing existing literature and trying to properly compare existing methods.
A model cascade is when a sequence of models (of growing computational cost and accuracy) is queried iteratively, but when the iteration can be stopped if a model's prediction is judged confident enough. This allows to trade off accuracy and cost in a natural way, and can be done with pretrained models without any need for finetuning.
The paper makes the following main empirical observations:
- Choosing models with large computational cost differences for the cascade induces the most cost-efficient trade offs
- The ideal exit strategy and ensembling strategy are problem and ensemble-choice dependent, but generally thresholding (with thresholds induces from the validation set) and taking the most confident prediction works
- 2-model cascades are already good, 3-model cascades only marginally improve performance


Generally the paper suggests that we can already do interesting things with very simple cascades, and so that we may be able to do better with more research on the topic.



**Audience:**

Yes

**Broader Impact Concerns:**

There is no Broader Impact Concerns statement in the paper.

**Claims And Evidence:**

Yes

**Requested Changes:**

This wouldn't be a review if it didn't ask for more baselines, and so I do wonder how this compares against dynamic models, either "early exit"-type models or even conditional-type models (e.g. large mixture of experts, like [1] and its descendants) that use a cheap model to route the input to the appropriate model.
I think this is an important baseline class because (a) it also allows for natural trade offs, and (b) in the "limit" of us getting better at ML I'd bet on the model class where the whole model is "aware" of its different parts over an amalgamation of agnostic models combined after the fact (and not necessarily "tuned" with each other).

[1] Shazeer, Noam, Azalia Mirhoseini, Krzysztof Maziarz, Andy Davis, Quoc Le, Geoffrey Hinton, and Jeff Dean. "Outrageously large neural networks: The sparsely-gated mixture-of-experts layer." arXiv preprint arXiv:1701.06538 (2017)

I'm in some sense surprised these methods work at all given the known lack of calibration of deep neural networks [2, 3]. I think it would make sense to spend a little more time on this (and perhaps even empirically justify the use of softmax logits almost as-is for iterative model selection -- this is the kind of "scientific investigation" I'm trying to allude to above).

[2] Guo, Chuan, Geoff Pleiss, Yu Sun, and Kilian Q. Weinberger. "On calibration of modern neural networks." In International conference on machine learning, pp. 1321-1330. PMLR, 2017.
[3] Wang, Deng-Bao, Lei Feng, and Min-Ling Zhang. "Rethinking calibration of deep neural networks: Do not be afraid of overconfidence." Advances in Neural Information Processing Systems 34 (2021): 11809-11820.


Some more comments:
- Maybe I'm missing something but on several occasion the authors write about "research that is currently missing." I'm not sure I see specifically what this is about; it would be helpful to expand on this.
- I'm not sure I understand the whole paragraph at the top of page 4. $A_2$ is introduced but never used, and then the grammar of the sentences following that is a bit weird. Would it be possible to rephrase?
- Also on page 4, after Eq (3), the NLL is minimized on what? A single example? A data set? I'm assuming the validation set?
- Figure 6, it would make sense if the Pareto front was shifted down by a constant factor (as much as the performance of the base models of the cascade is decreased by), but it could also be slightly better (the cascade could induce some robustness somehow). Is this the case? If not I'm not sure why this particular experiment is relevant in the paper. Are cascades normally known to _not_ be robust?
- When evaluating Pareto fronts, it's common to reduce them to some quantity so that different methods can be quantitatively compared. This can be done with indicators like the hypervolume, generational distance, and so on. Might be worth considering it here, since it does appear that methods in this paper do not strictly dominate one another.
- It's interesting that, similarly to ensembles, 2 or 3 models are sometimes plenty enough. I do wonder if this could be stretched further, and this is perhaps a part that's lacking in this paper. What could we do with 5, 10 models? Would this test the very idea of cascades? In some sense, if a model is too trivial, then we cannot trust its own self-confidence. What are the limits of that?
- Obviously the aim of this paper was not to do so, but it would be interesting to speculate more on how much better the trade off could be if models were trained _with the intent_ of using them in a cascade. Intuitively, the gap should exist, since it exists in e.g. pruned and quantized models.


**Strengths And Weaknesses:**

The paper is an investigative work. It summarizes past work and compares different methods on equal footing. Some analysis is done to understand what are the right design choices for this category of models, and to give us intuition on the method. By comparing lots of models and setups it also is a trustworthy data point.

While I appreciate this work, it feels more like a practical guide than a scientific investigation. I mean by that that it investigates the "how" (how do I make a good cascade) more than the "why" (what makes a good cascade good?). Generally I think this is where this paper could benefit the most from, unfortunately I don't have specific advice on what direction. I do think doing better on this front would help spur more precise and informative follow up work (something the authors seem to suggest is missing to the field).

I do have a number of comments on the work, see the following section, but I'm otherwise content with this being a TMLR paper. It was easy to read and understand, and appears to be empirically sound.

---

> ### Author Response · Authors · 2023-07-15
> **Response to Reviewer RnjH part 1**
>
> Thank you very much for the thoughtful feedback, we appreciate your interest!
>
> > This wouldn't be a review if it didn't ask for more baselines, and so I do wonder how this compares against dynamic models, either "early exit"-type models or even conditional-type models (e.g. large mixture of experts, like [1] and its descendants) that use a cheap model to route the input to the appropriate model.
>
> > I think this is an important baseline class because (a) it also allows for natural trade offs, and (b) in the "limit" of us getting better at ML I'd bet on the model class where the whole model is "aware" of its different parts over an amalgamation of agnostic models combined after the fact (and not necessarily "tuned" with each other).
>
> Unfortunately, in most works on early exit type models the code and models are not released, only small datasets such as MNIST or CIFAR are used, or important details are missing. Furthermore, results are usually far from the monolithic model Pareto front, thus practically not very meaningful. However, we have included a comparison to Table 2 of [4], which shows the results for a ResNet-50 on ImageNet for three different methods (Shallow-Deep Networks, Patience-based Early Exit, Zero Time Waste). Their accuracy drops by 5.7%-10.4% when average inference cost is reduced by 25% and continues to degrade more rapidly as inference cost is reduced further. At the model size which corresponds to ResNet-50 we obtain a 59% reduction in inference cost at no loss of accuracy for 3-model cascades. A more rigorous comparison would require replicating multiple works and then scaling up and optimizing their methods. This would require a great amount of resources and time.
>
> This is why we provide a baseline for consistent and rigorous evaluation in future work, to enable meaningful comparison and to show whether a method can beat state-of-the-art monolithic models.
>
> While it makes intuitive sense that learned pathing in a large dynamic model could be better, the added complexity can make such a model difficult to train. Cascades represent a complementary approach that can be combined with dynamic models and the simplicity of cascades is one of its key selling points. They provide large improvements in efficiency practically for free and can use any existing state-of-the-art models.
>
> [4] Maciej Wołczyk, Bartosz Wójcik, Klaudia Bałazy, Igor T Podolak, Jacek Tabor, Marek Śmieja, and Tomasz Trzcinski. "Zero time waste: Recycling predictions in early exit neural networks." Advances in Neural Information Processing Systems, 34:2516–2528, 2021.
>
> > I'm in some sense surprised these methods work at all given the known lack of calibration of deep neural networks [2, 3]. I think it would make sense to spend a little more time on this (and perhaps even empirically justify the use of softmax logits almost as-is for iterative model selection -- this is the kind of "scientific investigation" I'm trying to allude to above).
>
> It does not matter if the models are over- or under-confident. This will merely shift the threshold. What matters is the relative order of confidence scores for the predictions. Calibration does not necessarily improve this order.
>
> Cascades rely on two assumptions: (1) that the confidence score correlates with prediction accuracy, and (2) that the separate models make different predictions.
>
> 1: If the model was more confident at lower accuracy, it would tend to forward correct predictions and exit early with false predictions. However, a correlation between confidence and accuracy is a natural result of training with cross entropy loss, which nudges the model to be more confident about correct predictions and less confident about false predictions.
>
> 2: A simple example for correlated models is to use the same model twice in a cascade. In this case, the second model will never improve the prediction of the first model and the cascade will only add overhead. However, a difference in predictions is guaranteed in practice due to the randomness involved in the training process and is further increased by a difference in training data, architecture, and training method.
>
> Therefore, cascades achieve a convex trade-off, where accuracy tends to improve quickly as early exit rate falls and then starts to plateau around the accuracy of the best model. Figure 2 represents some of our first steps to investigate this.

---

> > ### Author Response · Authors · 2023-07-15
> > **Response to Reviewer RnjH part 2**
> >
> > > Maybe I'm missing something but on several occasion the authors write about "research that is currently missing." I'm not sure I see specifically what this is about; it would be helpful to expand on this.
> >
> > We have adjusted this statement in our paper. The point is that a baseline is currently missing (prior works we found typically investigate specific methods but don't compare themselves to methods from other works). The baseline we provide can be used for future research to enable a comparison between methods. If a new method is proposed, it can be evaluated on our baseline to see whether it is actually better than existing methods. We describe what is missing from prior work in the section about related work.
> >
> >
> > > I'm not sure I understand the whole paragraph at the top of page 4. $A_2$ is introduced but never used, and then the grammar of the sentences following that is a bit weird. Would it be possible to rephrase?
> >
> > We have rephrased this paragraph to explain how we compute improvement by comparing the inference cost of the baseline and cascade at equal accuracy.
> >
> >
> > > Also on page 4, after Eq (3), the NLL is minimized on what? A single example? A data set? I'm assuming the validation set?
> >
> > You are correct, it is minimized on the validation set. We have clarified this.
> >
> > > Figure 6, it would make sense if the Pareto front was shifted down by a constant factor (as much as the performance of the base models of the cascade is decreased by), but it could also be slightly better (the cascade could induce some robustness somehow). Is this the case? If not I'm not sure why this particular experiment is relevant in the paper. Are cascades normally known to not be robust?
> >
> > We are not aware of prior work which explores cascade robustness to distribution shifts. It is a concern that cascades may be vulnerable to distribution shifts because the smallest model makes the majority of decisions and fixed thresholds are used. To quote [3]: "the behaviour with distribution shifts might be most important in practice." Therefore we investigate this and show that while cascade performance degrades, they remain beneficial.
> >
> >
> > > When evaluating Pareto fronts, it's common to reduce them to some quantity so that different methods can be quantitatively compared. This can be done with indicators like the hypervolume, generational distance, and so on. Might be worth considering it here, since it does appear that methods in this paper do not strictly dominate one another.
> >
> > We compare average improvement across accuracy for some experiments in Section 4.1. We also considered various other indicators to compare the methods quantitatively, ranging from hypervolume to counting how many cascades work best for which method according to various criteria (such as average and best Pareto improvement, or how well a cascade maintains the accuracy of the best model). We felt that quantitative indicators do not add to the paper because they do not change the conclusion, are difficult to interpret, and the choice of indicator introduces a bias. Meanwhile, comparing the Pareto improvement shows how well each method performs at a glance and in a meaningful way. The takeaway should not be exact numbers (which change depending on model, dataset, and so on), but the trend that can be observed based on a large sample size for multiple experiments.
> >
> >
> >
> >
> > > It's interesting that, similarly to ensembles, 2 or 3 models are sometimes plenty enough. I do wonder if this could be stretched further, and this is perhaps a part that's lacking in this paper. What could we do with 5, 10 models? Would this test the very idea of cascades? In some sense, if a model is too trivial, then we cannot trust its own self-confidence. What are the limits of that?
> >
> > Our paper focuses on increasing inference efficiency in a practical way. There are many problems with larger cascades. The most important one is that efficient cascading relies on high early exit rates, typically around 70-80% after every model. Already for 3-model cascades, the last model will receive only about 5% of inputs. This results in diminishing returns for adding new models. A 10th model is expected to receive perhaps one in a million inputs, meaning its impact is close to zero.
> >
> > While we did not investigate the limits of cascade size, we demonstrate the trend of diminishing returns and believe that larger cascades are not practical. The following work investigates the effect of converting ensembles of 10 or 20 models to early exit cascades:
> >
> > Hiroshi Inoue. "Adaptive ensemble prediction for deep neural networks based on confidence level." In The 22nd International Conference on Artificial Intelligence and Statistics, pp. 1284–1293. PMLR, 2019.

---

> > > ### Author Response · Authors · 2023-07-15
> > > **Response to Reviewer RnjH part 3**
> > >
> > > > Obviously the aim of this paper was not to do so, but it would be interesting to speculate more on how much better the trade off could be if models were trained with the intent of using them in a cascade. Intuitively, the gap should exist, since it exists in e.g. pruned and quantized models.
> > >
> > > Indeed, our aim was to not conduct any training. We observe a trend that the best cascades contain models that were pretrained on different datasets and have different architectures (CNN and transformer). This aligns with the findings of works on ensembles such as [5]. Performance could be improved by decorrelating the models, for example with finetuning, or techniques similar to boosting. However, in practice, it is not trivial to train the models for a cascade. The most effective method to decorrelate models is to train on different data. However, the individual models will perform worse if they are trained on less data. The largest model requires the most training but also sees the fewest samples in the cascade.
> > >
> > > [5] Raphael Gontijo-Lopes, Yann Dauphin, and Ekin Dogus Cubuk. "No one representation to rule them all: Overlapping features of training methods." International Conference on Learning Representations, 2022.
> > >
> > >
> > >
> > > Please let us know your thoughts. We value your feedback!

---

> > ### Comment · Reviewer_RnjH · 2023-07-28
> >
> > Thanks for the response, clarifications and updates! I basically just have one follow up question:
> >
> > > a correlation between confidence and accuracy is a natural result of training with cross entropy loss
> >
> > Pardon my ignorance but is this a well known result? Is there empirical evidence for this somewhere?

---

> > > ### Author Response · Authors · 2023-08-01
> > >
> > > > Pardon my ignorance but is this a well known result? Is there empirical evidence for this somewhere?
> > >
> > > This is a great question. We are not aware of relevant literature. We mentioned the cross entropy loss since it is widely used for classification.
> > >
> > > Imagine accuracy $a_i$ is the true probability that $i$ is the correct class and $p_i$ is the predicted softmax. For a single example and correct class, the cross entropy loss gradient for class $i$ is $p_i-1$ if $i$ is the correct class and $p_i$ if it is not the correct class. The expected gradient is $a_i (p_i-1) + (1-a_i) p_i$. Let us rewrite $p_i = a_i + c$ so that the expected gradient becomes $a_i (a_i+c-1) + (1-a_i) (a_i+c) = c$. Therefore, the expected gradient can be interpreted as corrective force, which tries to align $a_i$ and $p_i$ and creates a correlation between confidence and accuracy.
> > >
> > > This is of course an oversimplification. In practice there are many other factors such as overfitting [2], regularization [3], or mismatch between training and deployment data. To what extent is the miscalibration a shift and to what extent is the order affected? The former is addressed by shifting the threshold but the latter is a concern.
> > >
> > > We examined the calibration of the models used in our ImageNet experiments (which can be done for example with reliability diagrams like in [2-3], or more finegrained with a moving average). Many models seem quite well calibrated, and the accuracy tends to increase with the maximum softmax for all models, meaning that stronger miscalibration appears to primarily be a shift.
> > >
> > > For ideal cascade performance we want a confidence score threshold above which every prediction is correct and below which every prediction is false. We tried to further investigate to what extent the order is affected by inspecting curves of model accuracy above and below respective thresholds representing various early exit rates, and temperature scaled calibration seemed to have little impact.

---

> > > > ### Comment · Reviewer_RnjH · 2023-08-01
> > > >
> > > > Thanks! If it's not already there I think it would be good to add this little investigation to the appendix, just to give the curious reader a bit more confidence that calibration was empirically analyzed and is a fair assumption.

---

> > > > > ### Author Response · Authors · 2023-08-03
> > > > >
> > > > > A section about calibration has been added at the end of the appendix (pages 18-20). We would appreciate it if you could take a look and let us know if there are any issues. Thank you!

---

### Decision · Action_Editors · 2023-08-19

**Recommendation:** Accept as is

**Comment:**

The reviewers recommend to accept this paper, after several points were clarified in the discussion with the authors. The reviewers comment that the paper asks its field to take a pause and think, rather than being a paper with a groundbreaking novel contribution. The reviewers feel this paper presents a straightforward yet effective approach for modeling cascades, striking an optimal balance between accuracy and efficiency. The paper is well written, the edits post-review are helpful, and the technical contribution feels interesting and relevant to move things forward in that subfield.

**Audience:**

The target audience is appropriate for TMLR

**Claims And Evidence:**

The claims made in the submission are supported by evidence satisfactorily